# Olfactory Stimulation as Environmental Enrichment for Domestic Horses—A Review

**DOI:** 10.3390/ani13203180

**Published:** 2023-10-12

**Authors:** Ana Caroline Bini de Lima, Vanessa Cristini Sebastião da Fé, Maria Simara Palermo Hernandes, Viviane Maria Oliveira dos Santos

**Affiliations:** Nucleus of Studies in Ambience, Bioclimatology, Welfare and Ethology, Faculty of Veterinary Medicine and Animal Science, Federal University of Mato Grosso do Sul, Campo Grande 79070-900, Brazil; vanessacristini.zoo@gmail.com (V.C.S.d.F.); mariasimarap@gmail.com (M.S.P.H.); viviane.oliveira@ufms.br (V.M.O.d.S.)

**Keywords:** animal welfare, behavior, equine, essential oils, odorants, odors, stress, smell

## Abstract

**Simple Summary:**

Equine welfare is a topic that has attracted the attention of both the scientific community and the public. In the domestic environment, horses are often kept in suboptimal conditions that limit the expression of species-typical behaviors and compromise their welfare. Environmental enrichment strategies have been identified as an interesting option to make the domestic environment more interactive and complex, resulting in an improvement in the animal’s quality of life. Olfactory stimulation, a form of sensory environmental enrichment that aims to trigger the sense of smell through the introduction of odors, has shown promise for horses. This review aims to present current information on equine olfaction, demonstrating its relevance as a sensory modality and exploring the potential that olfactory stimulation has as environmental enrichment for the species.

**Abstract:**

Horses constantly face several challenges inherent to the domestic environment, and it is common for the expression of their natural behavior to be drastically limited. Environmental enrichment has been suggested as an alternative to improve the captive situation of domestic horses. Among the recently proposed enrichment strategies, olfactory stimulation has emerged as a method for improving several aspects related to animal behavior. Olfaction is a sensory modality that plays a significant role in the expression of equine behavior, and in recent years, studies have shown that olfactory stimulation can influence the physiological and behavioral parameters of horses. This review provides current information on the anatomical particularities of the equine olfactory system, presents the physiological mechanisms involved in the odor detection process, and demonstrates how stress can interfere with this process. Finally, the use of olfactory stimulation as an environmental enrichment for domestic horses (*Equus ferus caballus*) is explored. The need for new studies that answer pertinent questions related to this topic is discussed throughout the manuscript.

## 1. Introduction

Domestic horses face several challenges inherent to the environment in which they are kept and managed. To deal with these challenges, animals rely on behavioral and physiological stress responses that, although adaptive, when provoked for a long time and/or frequently, can compromise their welfare [1,2].

Environmental enrichment has been suggested as an alternative to improve the captive situation of domestic horses and the safety of handlers [3,4]. In general, enrichment strategies involve providing sensory stimuli, new objects, social contacts, and the possibility of voluntary exercise [4], which results in an interactive and complex environment, allowing animals to express species-typical behavioral patterns. Olfactory stimulation is a form of sensory enrichment that holds considerable potential. Odors are relatively cheap, easy to store, and can become dynamic in time and space, rendering an aspect of novelty [5], which makes olfactory stimulation interesting from a practical point of view.

Horses have anatomical particularities indicative of a well-developed sense of smell, and thus the information obtained from olfactory stimuli is important for the species [6]. Recent studies have shown that olfactory stimulation can influence the physiological and behavioral parameters of domestic horses [7,8,9,10].

This review aims to present the current information on equine olfaction, demonstrating its relevance as a sensory modality and exploring the potential that olfactory stimulation has as an environmental enrichment strategy for this species.

## 2. Olfactory Perception in Horses

Odors form a key component of the environment and can provide crucial information for animals to interact with the environment effectively. An odor can be described as a mixture of volatile molecules (odorants) diluted in air or water, which are perceived by animals as an olfactory sensation [11,12].

Odor detection is a function of smell, one of the sensory modalities involved in chemoreception in mammals, allowing organisms to respond to chemical stimuli [13]. Through smell, external chemical information is transported and transformed in the central nervous system into patterns of brain activity [14].

The potential of a species to obtain olfactory information is influenced by nasal structure and breathing patterns [15]. Horses breathe exclusively through the nose, are capable of moving large volumes of air, and are among the terrestrial vertebrates with the highest number of olfactory receptors [15,16,17]. Furthermore, through the action of sniffing, they are able to intensify the air current in the nasal passages, which results in more intimate contact between odorants and olfactory receptors [18].

During inspiration, the odor carried by air enters through the nostrils, which in horses are separated, oriented in different directions, and have fine and short hairs that act in filtering the air. Next, the air enters the nasal passages, which consist of a pair of nasal cavities divided by the nasal septum that extends 20 to 30 cm [19].

The nasal cavities are lined by the nasal mucosa and covered with a ciliated respiratory epithelium; this mucosa includes serous glands, whose watery secretions hydrate and warm the incoming air, as well as a dense vascular network that adjusts the temperature to that of the body [13]. Thus, the cavities modify the properties of the inhaled air before reaching other parts of the respiratory tract, exercising a thermoregulatory function.

The horse’s nasal cavities contain two tightly rolled turbinate bones that force air to circulate over as much mucosal surface area as possible [20]. The turbinates divide each nasal cavity into the dorsal, middle, common, and ventral meatus [19]. Through the dorsal meatus, inspired air is transferred to the ethmoid area, located in the caudal part of the nasal cavity. In this area, the ethmoid turbinates are located, which, in this species, are small but numerous, increasing the olfactory surface [21] (Figure 1).

According to Kupke et al. [22], the equine olfactory epithelium is predominantly located in caudodorsal areas of the nasal turbinates with a significant decline in the rostroventral direction, especially for type a epithelium. In this work, two epithelial subtypes were identified, designated as types a and b, which resemble those previously described in dogs. Type a resembles mature epithelium, in contrast to the more juvenile type b.

In general, the olfactory epithelium is composed of olfactory cells (olfactory receptor neurons), supporting cells, and basal cells [22] (Figure 1). While the supporting cells surround the olfactory cells, ensuring a stabilized ionic environment, the basal cells divide and differentiate into supporting cells or olfactory cells, ensuring the cell renewal of the olfactory epithelium [23,24].

Olfactory cells are bipolar neurons composed of a cell body, a dendrite, and an axon [22] (Figure 1). At the dendritic end of these neurons, there is an olfactory knob that gives rise to olfactory cilia that spread across the surface of the olfactory mucosa [22]. These cilia are surrounded by mucus that limits the toxicity of the environment and plays an important role in making the lipophilic odorant molecules more soluble in this hydrophilic gel [23], which is essential for these molecules to reach an olfactory receptor capable of recognizing them.

Odorant molecules interact with the G-protein-coupled receptors (GPCRs) present in the membrane of olfactory cilia. An odorant can activate several receptors, and each receptor is capable of detecting more than one odorant. In this way, different odors are represented as different combinations of activated receptors. This mechanism of combinatorial receptor coding is used by the olfactory system of mammals to identify and discriminate odorants [17,25].

Olfactory signaling is initiated when odorants bind to receptors, activating a signaling pathway that produces an intracellular messenger. This biochemical signal is transduced into an electrical signal by the opening of ion channels [26]. The generated action potentials propagate along the axons of the olfactory neurons that form olfactory nerve fibers (cranial nerve 1—CN I) and terminate in the olfactory bulb, where the terminal ends of the olfactory fibers synapse with the dendrites of tufted and mitral cells, forming the glomeruli of the olfactory bulb [27].

The olfactory bulb is the first site of olfactory processing in the central nervous system [28]. Recently, in a pilot study comparing the skull morphology of horses and donkeys, Merkies et al. [29] observed that the hair whorl on the forehead of horses almost always corresponded with the location of the olfactory bulbs. In addition, the olfactory bulbs of horses were larger than those of donkeys.

Axons from mitral and tufted cells leave the olfactory bulb, forming a large lateral olfactory tract, and thus transmit information to neurons in the primary olfactory cortex, which redirect olfactory information to various brain structures involved in memory, learning, and emotions (hypothalamus, thalamus, amygdala, hippocampus, and orbitofrontal cortex) [13,27].

Unlike other sensory modalities, olfactory information ascends ipsilaterally from the detection area located in the nasal cavity to the perception area in the brain [30]. In previous research, Siniscalchi et al. [31] observed that horses showed a preferential pattern in the use of the right nostril when sniffing samples of adrenaline and urine, which suggests the involvement of the right hemisphere in the analysis of intense emotions and sexual behavior.

In addition to the main olfactory system, the horse has a well-developed accessory olfactory system (vomeronasal organ—VNO), sensitive to non-volatile and low-volatile particles, which are common in body secretions [6,21]. The vomeronasal organ consists of a pair of ducts with a tubular structure, surrounded by cartilage and located at the base of the nasal septum [32,33] (Figure 1).

At the caudal end, the vomeronasal ducts are closed, but at the rostral end, they open into the incisive canals, which in several species of mammals (including dogs, cattle, and pigs), connect the nasal and oral cavities through openings in the hard palate [21,33,34]. However, in horses, there is no communication between the VNO and the oral cavity, maintaining only the usual connection with the nasal cavity [33].

The vomeronasal ducts are lined by sensory and non-sensory epithelia. The non-sensory vomeronasal epithelium is found in the lateral portion of the VNO and is composed of ciliated cells and basal cells. The sensory vomeronasal epithelium is found in the medial portion of the VNO and is composed of receptor cells (vomeronasal sensory neurons), supporting cells, and basal cells [32].

VNO receptor cells are classified into two types: cells that express vomeronasal receptor type 1 (V1R) and cells that express vomeronasal receptor type 2 (V2R) [35]. In horses, it was observed that the VNO receptor cells were positive for Gαi2 protein (V1R marker) but not for Gαo (V2R marker) [32]. Therefore, in this species, as also observed in dogs and cattle [36], only V1R is expressed.

Upon the binding of a molecule to a V1R receptor, the activation of the G-protein triggers biochemical cascades that result in ion channel activation and a depolarizing transduction current. The action potentials generated are propagated along the vomeronasal nerve to the accessory olfactory bulb [26,37]. In animal species that have only V1R, receptor cells that express V1R project axons over the entire glomerular layer of the accessory olfactory bulb [35]. Vomeronasal signals are processed in the accessory olfactory bulb and then directly relayed to the amygdala, where there are connections to hypothalamic neuroendocrine centers [37].

The activation of the vomeronasal organ appears to be facilitated by the flehmen response. When expressing the flehmen response, the horse extends its neck, lifts its nose, opens its mouth slightly, and lifts its upper lip [38]. During the expression of this behavior, air leakage is reduced, which results in increased air pressure within the nasal cavity, allowing the horse to analyze low-volatile compounds with greater accuracy [6] (Figure 2).

In the past, the main and accessory olfactory systems were seen as separate pathways, involved in detecting distinct sets of olfactory cues [39]. It was believed that the main olfactory system was exclusively responsible for the perception of volatile odorants, while the accessory olfactory system was responsible for the perception of non-volatile pheromones [40]. However, more recent research indicates that the two olfactory systems act in a complementary way [39,41], and even some molecules are detected by both, as demonstrated by Choi et al. [42]. In this study, it was observed that thoroughbred horses are capable of androstenone (steroid pheromone) perception through an odorant receptor (OR7D4) expressed in the VNO and the olfactory epithelium.

## 3. Olfactory Perception in Relation to Stress

There is a wide range of stressors to which domestic horses are often exposed. The domestic environment can be extremely challenging, especially when horses are kept in restrictive systems and used in physically demanding activities. In this context, horses may face internal and external stressors, including painful stimuli, fear-eliciting stimuli, and lack of satisfaction with biological needs, as well as psychological ones, which include the loss of control and predictability [1].

In the face of a potential stressor, the biological defense coordinated by the central nervous system consists of a combination of behavioral, autonomic, and neuroendocrine stress reactions [1]. The altered sense of smell may be part of this coordinated response. Several regions of the brain associated with olfactory responses, including the epithelium and the olfactory bulb, express receptors for adrenaline, CRH, or glucocorticoids (stress hormones), which reflects its response potential to these hormones [43]. Thus, the perception of odor can be altered during its transmission from the peripheral nervous system to the brain due to stress and associated emotions, such as fear [11].

This change in perception appears to differ between acute and chronic stress conditions. While acute stress seems to modify the salience of olfactory stimuli and alter their perception and memorization through general arousal and specific olfactory mechanisms, chronic stress, often associated with depressive-like symptoms, is related to losses in olfactory perception [43]. Given this information, it should be noted that the domestic horse was considered a potentially useful candidate for an animal model of depression [44].

If, on the one hand, the olfactory perception is altered by the stress response, on the other hand, the very odors involved in animal chemical communication can signal danger and, thus, induce stress in recipient animals [11]. The odor of predators, as well as the odor of stressed conspecifics, can be perceived as aversive by farm animals [45]. Christensen et al. [46] reported that horses exposed to the blood of stressed conspecifics and predator odor showed significant behavioral changes, including increased vigilance behavior, although no significant increases in heart rate were observed.

In the domestic environment, the odor of stressed conspecifics deserves attention, since there is a great chance that animals are exposed to these compounds. The chemical information transmitted through the odor of stressed conspecifics provides an evolutionary advantage, alerting other individuals of the same species about possible threats. This facilitates a process called emotional contagion, which prompts animals to shift their affective state in the same direction when perceiving a certain emotional state [11,47].

Although stress reactions are adaptive under conditions in which the animal is exposed to several stressors or the same stressor repeatedly, they may still pose possible harmful effects on biological functions, because the organism cannot replenish its biological resources, thus accumulating biological costs, which negatively affects animal welfare [48]. In addition to the clear impact that stress can have on the organism, handling stressed animals can be riskier, as stress-related behaviors (agitation and aggression, for example) can pose a danger to people involved in the care and training of horses. In this context, another factor that must be considered is the odor of people in contact with these animals, since it has already been demonstrated that the olfactory information obtained from human body odor alone is sufficient to induce some differential behavior in horses [49].

There is an active search in the scientific community for alternatives that can reduce stress. The use of odors as stress relief agents has been investigated and questioned, especially due to the factors related to the methodology adopted in some experiments [11]. There are gaps in knowledge that should be the target of future research; however, it is important to highlight that research carried out with horses so far has shown promising results [8,50,51].

## 4. Olfactory Stimulation as Environmental Enrichment

Environmental enrichment can be defined as an improvement in the biological functioning of captive animals resulting from modifications in their environment [52]. The practical implementation of environmental enrichment is achieved through strategies that, in general, aim to increase behavioral diversity, reduce the frequency of aberrant behaviors, increase the variety or number of species-typical behavioral patterns, increase the positive use of the environment, and the ability of animals to cope with challenges [53].

In recent years, sensory stimulation, a practice based on the use of stimuli designed to trigger one or more of the animal’s senses, has been explored as a method of environmental enrichment [54]. Although olfaction is a sensory modality neglected by applied ethology and animal welfare [5], research carried out recently with several domestic species, using odors from animals or plants as olfactory stimuli, has shown increasing interest in the study of the effect of olfactory stimulation on behavior [9,55,56,57].

Olfactory stimuli can be presented to animals in different ways. According to Clark and King [58], odor presentations can be classified as concentrated, semi-concentrated, and dispersed. While the concentrated presentation involves providing the odor in or on a receptacle (cloth, cups, and logs), the semi-concentrated presentation involves depositing the material that has the odor inside a sack, or other similar receptacle, which is usually ripped by the animals, causing the odor to be dissipated throughout the enclosure via physical action. The dispersed presentation, on the other hand, refers to filling the enclosure with odor; this can be carried out by applying it to the various items present in the enclosure, or by scenting the air through diffusers or sprays.

Concentrated and semi-concentrated presentations allow animals to have greater autonomy when interacting with olfactory stimuli, whereas dispersed presentations do not offer individuals the opportunity to move away from the odor (Figure 3), which must be considered when using potentially aversive odors [58].

The success of olfactory stimulation as a method of environmental enrichment may depend on the species under analysis [54]. For horses, a species in which chemical communication is a fundamental part of their social interaction [59], and which has anatomical features indicative of a particularly well-developed sense of smell [60], olfactory stimulation holds considerable potential.

### 4.1. Biologically Relevant Odors

Odors originating from the body, urine, or fecal matter from conspecifics and heterospecifics can be considered biologically relevant [54,61]. These odors often provide the first clues to biologically important stimuli, including approaching group members, mating partners, and predators. In addition to their high valence for survival, these olfactory stimuli are characterized by frequently inducing both innate and learned behavioral responses [62]. Therefore, the use of these odors as olfactory enrichment deserves attention.

In the study conducted by Guillaume et al. [7], concentrated presentation of urine from conspecifics induced a high level of olfactory investigation and flehmen response in stallions. The results demonstrated that stallions seem to not show differences in flehmen response between estrous mare urine, anestrous mare urine, and gelding urine. It has been observed, however, that the flehmen response expressed by stallions is greater after sniffing stallion urine than after sniffing gelding urine. In contrast to the responses observed in stallions, geldings when exposed to urine from conspecifics expressed a low level of olfactory investigation that did not differ with the type of urinary stimulation. Although geldings also expressed flehmen response, its occurrence was considered by the authors to be a very rare event.

When evaluating the impact of olfactory stimulation on sexual behavior, Guillaume et al. [7] showed that, although no differences were observed regarding the duration and number of flehmen responses according to the type of urine used (estrus mare vs. anestrus mare), the latencies of erection, mounting on the dummy and ejaculation were significantly reduced when stallions were stimulated by urine from a mare in estrus. Thus, both types of urine induce high levels of the flehmen response, but the sexual response varies considerably, with estrus urine inducing a stimulatory response, and anestrus urine inducing an inhibitory response on sexual behavior.

In the work carried out by Hothersall et al. [63], the presentation of concentrated urine from conspecifics to pregnant mares induced investigative responses, which consisted of sniffing and occasionally flehmen responses. Mares appeared to be able to discriminate between urine samples from other pregnant mares, and from geldings, suggesting that the urine contains some information that can be used to discriminate between classes of conspecifics. Furthermore, the authors revealed that mares seemed much more interested in investigating samples when individually housed during the winter when opportunities for interaction with conspecifics were limited.

Fecal odor also seems to play a significant role in different behaviors expressed by horses. When exposed to fecal samples from stallions and mares, domestic stallions sniffed mare feces longer and expressed more flehmen response [64]. In this experiment, although marking by defecation in the feces of stallions and mares did not differ, marking by urination did. Stallions urinated exclusively on mare feces and did not differentiate between estrus and diestrus mares’ feces. Marking behaviors are part of the behavioral repertoire of males, and it is plausible that, when defecating in male feces, stallions announce their presence to rivals, while when urinating in female feces, they mask the presence of mares, avoiding the interest of potential rivals [64].

The restriction of contact between mares and stallions in the domestic environment drastically limits the expression of their natural behavior [59]. The use of olfactory stimuli in this context could increase the variety of species-typical behavioral patterns (olfactory investigation, flehmen response, and marking). Domestic stallions are highly reactive to mares’ urine and feces; however, it should be investigated whether olfactory stimulation can generate frustration in males who will not have the opportunity to express mounting responses later on.

Concerning female exposure to fecal samples, Hothersall et al. [63] observed that pregnant mares showed no consistent response or evidence of discrimination between the feces of other pregnant mares and geldings. Krueger and Flauger [65] reported that mares, as well as geldings, were highly motivated to smell conspecific feces. In this study, the animals always paid attention to the samples they were confronted with, but they spent more time smelling the feces of conspecifics than their own. The results obtained by the authors also demonstrated that mares and geldings paid more attention to the feces of horses from which they received the greatest amount of aggressive behavior.

As conspecific odors can elicit olfactory investigation in castrated males and females, the use of this type of stimulus can also be interesting for these categories, especially in the case of social isolation. In this way, chemical communication, which is of extreme relevance for this species [59], can be facilitated even in circumstances where total physical contact is limited.

It is important to emphasize that the use of biologically relevant odors involves the transmission of chemical information; therefore, the identification of the sender and receiver of this information becomes essential for the success of olfactory stimulation and for avoiding undesirable results. Avoiding the use of odors from stressed conspecifics, as well as conspecifics that have engaged in agonistic interactions with the animal to which the odor will be exposed, can be an important measure to prevent a possible stress response from being induced in the receiver.

In addition to the odors of feces and urine, the body odor of conspecifics is relevant for horses in different social contexts. Even a common form of greeting among them, whether familiar or not, consists of sniffing nose–nose and nose–body [59]. In the work carried out by Hothersall et al. [63], mares’ responses to a cloth rubbed on another horse’s coat were extremely variable, but the relatively high average duration of olfactory investigation led the authors to recommend this type of sample to be tested in future experiments. Guarneros et al. [59] suggested that the odor of partners may help keep horses calm during transport periods. This suggestion also needs to be investigated.

When considering the use of biological materials as olfactory stimuli, some important points need to be considered, such as the presence of pathogens and parasites. According to Clark and King [58], testing for pathogens and parasites and exposing samples to extreme temperatures can decrease the risk of disease transmission.

An option that has been explored in some publications and does not involve biological material is the use of a synthetic analog of the equine-appeasing pheromone. Like several species of mammals, nursing mares produce a calming pheromone, which consists of a chemical message that provides reassurance for their foal, helping them to feel safe, protected, and more confident in new situations and environments [66]. Studies that aimed to evaluate the effect of the synthetic analog of the equine-appeasing pheromone in animals submitted to stressful conditions have revealed inconsistent results so far.

In the study developed by Falewee et al. [51], reduced fear responses were observed in adult horses given the synthetic analog of the equine-appeasing pheromone before a fear test. Alves de Paula et al. [67] evaluated untamed foals undergoing hoof trimming for the first time, and no significant behavioral or physiological changes were observed between the group that received the synthetic analog of the equine-appeasing pheromone, and the group that received a placebo. Berger et al. [68], working with abruptly weaned foals, reported that there was no significant effect of pheromone treatment on cortisol values or stress-related behavioral responses. In addition to the conflicting results, it is necessary to point out that, in these studies, the synthetic analog of the equine-appeasing pheromone was applied intranasally. For its use as an environmental enrichment strategy to be tested, in future experiments, the pheromone should be presented in a concentrated, semi-concentrated, or dispersed form.

The benefits of introducing odors of natural predators are uncertain, and even though considerable harmful effects have not been observed in domestic horses exposed to predator odors [46], the use of this type of stimulus as environmental enrichment is questionable and deserves to be investigated in future studies. Furthermore, according to Wells [54], the use of harmless, non-stressful stimuli in enrichment programs is likely to result in greater welfare benefits.

When choosing an olfactory stimulus, one should consider the fact that the interest in the odor of a heterospecific or conspecific will be affected by the familiarity of the animal, that is, their previous experience with the odor, and by the motivation to investigate it [58]. The sexual experience, for example, makes the animal more responsive to olfactory stimuli from females, increasing the efficiency of these stimuli in inducing the sexual response [7]. Another point to be observed is the occurrence of habituation to these olfactory stimuli, as the intensity of sniffing social cues decreases with the frequency of presentation [63,65]. 

### 4.2. Biologically Irrelevant Odors

Although not considered biologically relevant for many animals, essential oils and other plant-derived odors may be able to improve the welfare of certain species, according to the results of recent work [54]. While some of these odors seem to encourage relaxation and relieve stress, others seem to have a more stimulating effect on animals [56,69,70]. 

Plants have chemical components produced as part of their metabolism that can be classified as primary or secondary metabolites. Primary metabolites, such as sugars and lipids, are found in all plants, while secondary metabolites are found only in some genera or species, as they are not essential components for plant metabolism [71]. Secondary metabolites are used as plant defense mechanisms against microorganisms, insects, and herbivores, and also contribute to specific odors, flavors, and colors in plants [72]. Essential oils are volatile compounds, formed by aromatic plants as secondary metabolites, which are usually obtained via steam or hydrodistillation [73]. 

Most of the research carried out with horses focused on the use of lavender essential oil. This essential oil, known for its anxiolytic-like effect in animal models [74], has been shown to have the potential to reduce the intensity of stress reactions in horses according to recent studies [8,10,50]. The use of this type of olfactory stimulus has the potential to help animals deal with challenges posed by the domestic environment and reduce unwanted behaviors.

According to Ferguson et al. [10], horses treated with lavender essential oil (*Lavandula augustifolia*) after an acute stress response induced by an auditory stimulus (air horn) showed a significant reduction in heart rate, while Heitman et al. [8] reported that, in horses treated with lavender essential oil (*Lavandula augustifolia*) during transport cortisol, the levels were suppressed.

In the experiment conducted by Poutaraud et al. [50], horses were subjected to a series of stress tests in which they were exposed to common stressors in the domestic environment, such as social isolation, novel area, novel object, and the sudden appearance of an object. In this study, stress indicators such as heart rate, alert postures, and defecation were lower in horses treated with lavender (*Lavandula augustifolia*) before stress tests. The lavender essential oil also modified salivary cortisol. The pharmacokinetics of linalool in plasma peaked 20 min after the application of lavender essential oil, thus confirming its effect. However, a differential of this work was the dilution of the essential oil in vegetable oil and the application with the aid of a roll-on around the nostrils of the treated animals. From the point of view of environmental enrichment, this type of application is not ideal, as it does not constitute a change in the environment itself. Furthermore, the use of roll-on forces the animal to smell the odor, not allowing individuals to have autonomy while interacting with the olfactory stimulus.

In horses that were not subjected to an external stressor, Baldwin et al. [75] observed that the use of lavender essential oil (*Lavandula augustifolia*) significantly increased the parasympathetic component of heart rate variability (RMSSD). This effect was not maintained after lavender removal, which highlights that its use should be considered as a short-term solution. The researchers also tested the effect of chamomile essential oil (*Matricaria Recutita*); however, chamomile did not show a consistent relaxing effect on the autonomic nervous system.

More recent research has begun to explore the effect of a wider range of essential oils. Kosiara and Harrison [76] tested the calming effect of the essential oils of vetiver (*Vetiveria zizanioides*), spikenard (*Nardostachys jatamansi*), and roman chamomile (*Anthemis nobilis*) in relation to water (negative control) and lavender essential oil (*Lavandula angustifolia*, positive control) in horses standing still. The authors concluded that spikenard and Roman chamomile appeared to be better at calming horses than lavender, which has been the best-documented essential oil to date. While spikenard was better at inducing a relaxed facial expression, Roman chamomile was better in terms of muscle relaxation.

Unlike other studies, whose focus was on testing the ability of essential oils to encourage relaxation and relieve stress, the study conducted by Rorvang et al. [9] aimed to test the olfaction of horses that were presented with the odors of the following oils: orange (*Citrus sinensis*), peppermint (*Mentha piperita*), cedarwood (*Cedrus*), and lavender (*Lavandula angustifolia*). Their results demonstrated that the horses were able to detect and distinguish between all four odors. Young horses sniffed cedar longer than old horses and pregnant mares sniffed less lavender than non-pregnant mares. Furthermore, animals showed greater interest in peppermint essential oil, which calls for further investigation. In this experiment, behaviors indicative of averseness were low.

The work of Rorvang et al. [9] provided insights of extreme relevance for the choice and use of essential oils as olfactory enrichment, as it indicates a low aversion of horses toward these odors, even highlighting that some of them may be more or less suitable according to age and physiological state. Another point worth mentioning is that, while most experiments testing essential oils performed dispersed presentations [8,10,75,76], in this experiment, a concentrated presentation was performed, which allowed researchers to observe the interest of animals concerning odors. The animals’ response to peppermint essential oil also deserves attention because, unlike the others tested, this oil is known for its stimulating effect, having already been observed in other species [77]. 

The introduction of odors with stimulating properties can be advantageous for some species kept in captivity for long periods, helping in mental stimulation and psychological well-being. However, it is necessary to consider the fact that animals that exhibit a state analogous to depression may develop a more active type of aberrant behavior after exposure to stimulating odors [54]. The use of odors that cause greater agitation or stress is questionable, especially in the case of horses, which are often housed in places that restrict their movement and are routinely exposed to several stressors. Still, further research should explore the use of this type of olfactory stimulus in horses so that its impact on behavior is better understood.

## 5. Conclusions

Equine olfaction, although still little explored compared to other sensory modalities, is relevant for this species in several contexts. Research carried out to date has shown that several components of the equine behavioral repertoire can be influenced by olfactory stimulation. Olfactory stimuli have the potential to increase behavioral diversity and the ability of animals to deal with challenges, as well as to stimulate species-typical behaviors.

It should be noted, however, that there are still issues related to this area of knowledge that need to be addressed. The impact of different forms of odor presentation, the ideal interval between the introduction of different odors, the influence of the habituation process, long-term effects, and possible health risks (e.g., toxicity and the transmission of pathogens and parasites) are topics that deserve investigation.

When using olfactory stimulation as environmental enrichment, professionals working with horses should seek to use olfactory stimuli that have already been tested in scientific settings and are considered safe and effective for this species. Furthermore, olfactory stimuli must be used in contexts similar to those in which they were tested in experiments (e.g., lavender essential oil for stress reduction). This can be achieved based on the scientific studies presented in this review.

A stimulating environment can improve animal welfare, and olfactory stimulation can be a useful tool to enrich the environment of domestic horses.

## Figures and Tables

**Figure 1 animals-13-03180-f001:**
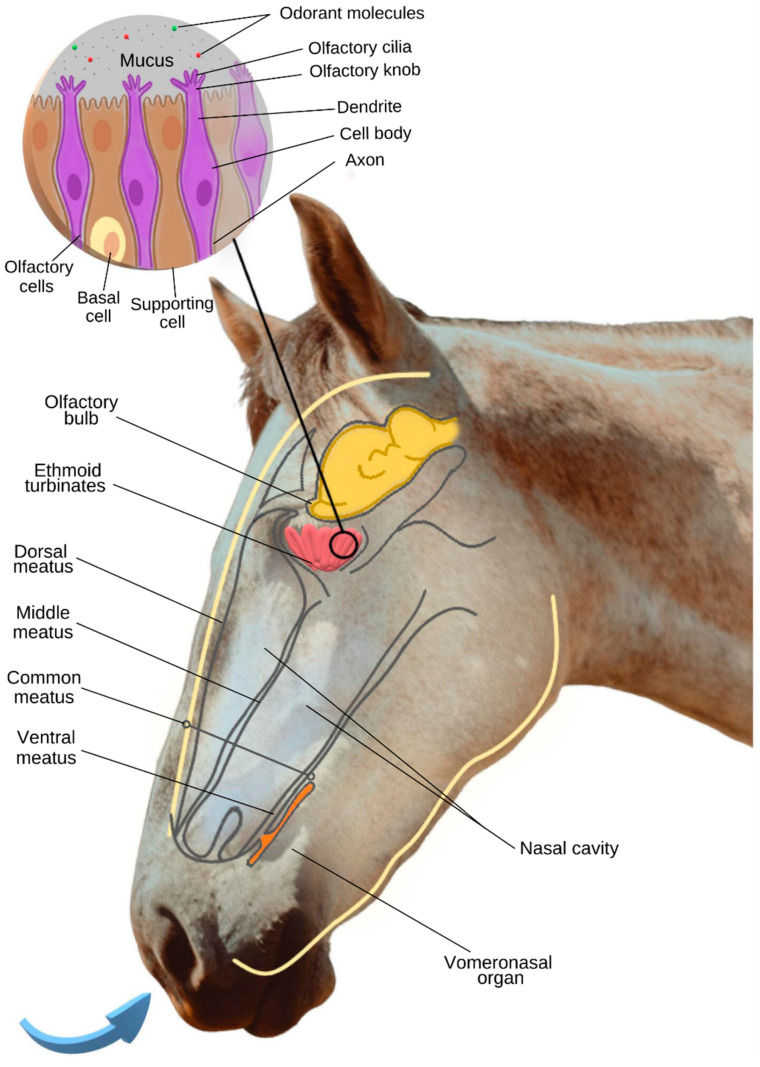
Overview of the structures and cells associated with the equine olfactory system. Information based on [6,20] (photo credit and illustration: A.C. Bini de Lima).

**Figure 2 animals-13-03180-f002:**
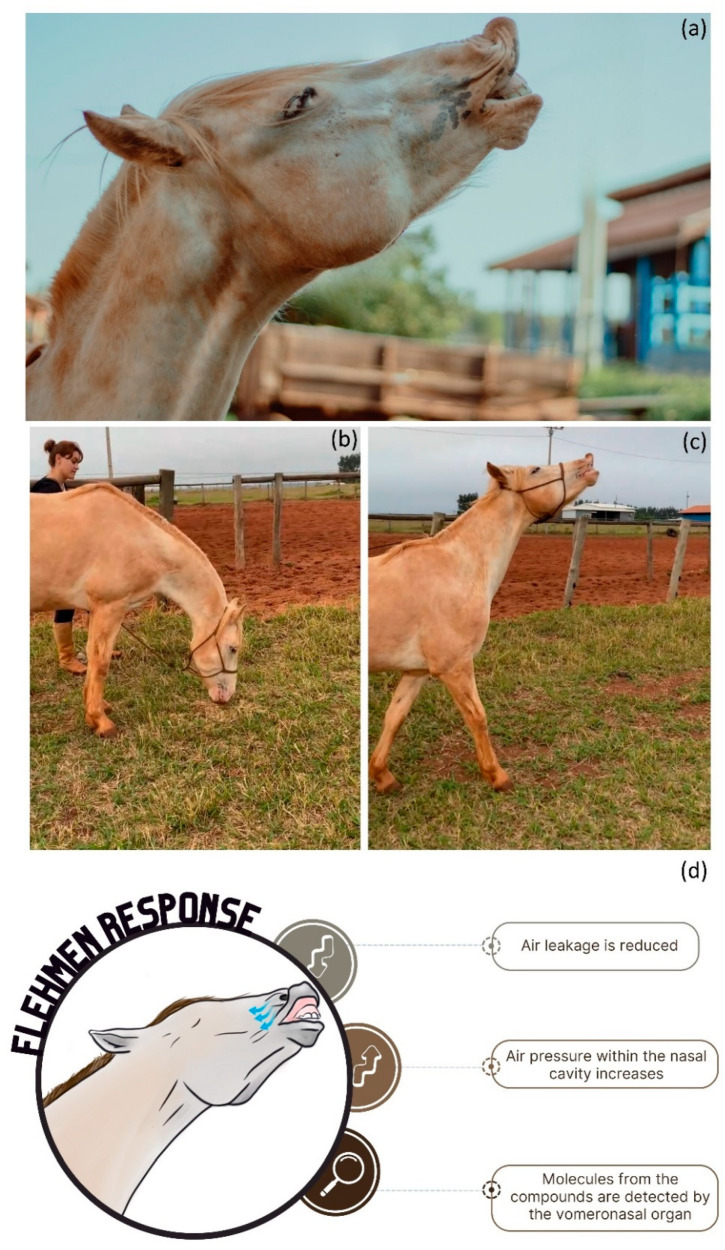
Equid expressing flehmen response: (**a**) stallion expressing flehmen response after olfactory stimulation with peppermint essential oil (photo credit: M.S.P. Hernandes); (**b**,**c**) stallion displaying olfactory investigation followed by flehmen response after being exposed to mare feces (photo credit: M.S.P. Hernandes); (**d**) infographic of the processes involved during the expression of the flehmen response, information based on [6] (illustration: A.C. Bini de Lima).

**Figure 3 animals-13-03180-f003:**
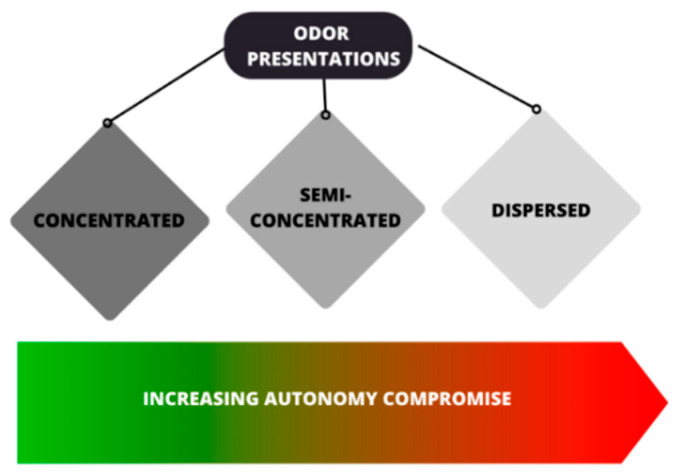
Diagrammatic representation of the classes of odor presentations, where the compromise of the animal’s autonomy while interacting with the olfactory stimulus increases according to the type of presentation. Information based on [58] (illustration: A.C. Bini de Lima).

## Data Availability

Not applicable.

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
