# Peer review of "Olfactory Stimulation as Environmental Enrichment for Domestic Horses—A Review"

_animals, 2023, doi:10.3390/ani13203180_

Round 1

Reviewer 1 Report

Dear Authors

The article proposed for review on "Olfactory stimulation as environmental enrichment for domestic horses - A review" is written in an interesting and correct way. The authors correctly described the anatomical elements that make up the structure of the sense organ, which is the organ of smell. They also described the significant role of pheromones in the studied species and the method of interpreting volatile odor substances. The article contains a lot of information well-known in the literature regarding the functioning of the sense being examined, but the fact that it is a review work in which much current knowledge from numerous literature items was used means that publication of this article can be considered. My minor comment to the manuscript: please post a photo (not a drawing) of the horse expressing the flehmen response - Fig. 2.

Regards

Author Response

Dear Reviewer,

Thank you very much for your contribution, we really appreciate every observation. The choice to use an illustration to represent Flehmen's response was made because it is common to use illustrations to represent behavioral responses in ethograms and because it attracts the attention of readers, however, considering your observations we will attach a photo as well, to further improve our review.

Kind regards

Reviewer 2 Report

Comments for authors: attached PDF file   

Author Response

Dear Reviewer,

Thank you very much for your contribution, we appreciate every observation.

Specific comments reply:

1 -  To underline the aim of the article is a great observation. We will promptly replace it based on your suggestion.

2 - 170 line: The information provided in the figures is based on literature and illustrations made by the authors, we will place this information in each of the figures to ensure transparency for readers.

3 - 191-194 line: We will promptly replace it based on your suggestion.

4 - 211-215 line: We will promptly replace it based on your suggestion.

5 - Great observation. The habituation process will be mentioned in the conclusion as suggested.

Kind regards

Reviewer 3 Report

This is a very well-written and comprehensive review of the role of olfaction in horses. The authors have exhausted the known information on olfaction in horses and provided very clear and relevant suggestions for future research. The only comment that perhaps would make the manuscript more salient is to spend a bit of time explaining how odor can be used as environmental enrichment (EE). The authors have explained what EE is, and how to achieve it, but not really touched on how/when/where one would use odor as EE and to what effect (other than calming instances, but this is not explicitly EE). This is especially important since this is a main point in your conclusion and abstract.

L415 – can you more clearly explain why a roll-on application of odor is controversial and provide a reference?

Some other minor formatting comments:

Throughout the manuscript the authors sometimes refer to the horse as “it”. While is it accepted that horses are sentient creatures, it would be more acceptable to refer to them as “them/they/their”.

L171 – note that “equine” is an adjective while “equid” is a noun. Thus the figure caption should read “Equid expressing flehmen response”

L401 and L403 the references appear to be in a different font

L489 – the authors in reference 4 are incorrectly cited. The last author should read Mosian, M-P.

L624 – the last names of authors in reference 72 should only have the first letter capitalized

well written with only minor editing required

Author Response

Dear Reviewer,

Thank you very much for your contribution, we appreciate every observation.

1- Our review aims to explore the potential that olfactory stimulation has to be used as environmental enrichment. To describe how/when/where to use it questions related to this topic raised and discussed throughout the manuscript would need to be answered.

Therefore, this type of specific recommendation would be the next step dedicated to future publications. Here in this review, our focus is to show the potential of olfactory stimulation and gaps related to this topic in the scientific literature so that we can move forward in this direction.

That said, as an initial step and for safety reasons we can include as a topic in our conclusion the recommendation that when using olfactory stimulation, professionals working with horses should seek to use olfactory stimuli that have already been tested in scientific settings and have been shown to be safe and effective for this species. Furthermore, olfactory stimuli should be used in contexts similar to those in which they were tested in the experiments (e.g., lavender essential oil for stress reduction). The scientific studies presented in this review can be used as a reference for this practice.

2 - Roll-on application is not ideal, as it does not constitute a change in the environment itself. Furthermore, the use of the roll-on forces the animal to smell the odor, not allowing individuals to have autonomy while interacting with the olfactory stimulus. This whole explanation will be included in the review and "not ideal" may be a better choice of words than "controversial" since this topic was not widely discussed to be considered controversial.

3 - Referring to the horses as “them/they/their” is a great observation. We will promptly replace it based on your suggestion.

4 - L171: We will promptly replace it based on your suggestion.

5 -  L401 and L403: We will promptly adjust them based on your suggestion.

6 - L489: We will promptly replace it based on your suggestion.

7 - L624: We will promptly replace it based on your suggestion.